# Flexible Piezoresistive Polystyrene Composite Sensors Filled with Hollow 3D Graphitic Shells

**DOI:** 10.3390/polym15244674

**Published:** 2023-12-11

**Authors:** Nataliia Guzenko, Marcin Godzierz, Klaudia Kurtyka, Anna Hercog, Klaudia Nocoń-Szmajda, Anna Gawron, Urszula Szeluga, Barbara Trzebicka, Ruizhi Yang, Mark H. Rümmeli

**Affiliations:** 1Centre of Polymer and Carbon Materials, Polish Academy of Sciences, M. Curie-Skłodowskiej 34, 41-819 Zabrze, Poland; guznataliia@gmail.com (N.G.); kkurtyka@cmpw-pan.pl (K.K.); ahercog@cmpw-pan.pl (A.H.); knocon@cmpw-pan.pl (K.N.-S.); agawron29@wp.pl (A.G.); uszeluga@cmpw-pan.pl (U.S.); btrzebicka@cmpw-pan.pl (B.T.); 2Chuiko Institute of Surface Chemistry, National Academy of Sciences of Ukraine, General Naumov Str. 17, 03164 Kyiv, Ukraine; 3International Polish-Ukrainian Research Laboratory ADPOLCOM, 41-800 Zabrze, Poland; 4Faculty of Biomedical Engineering, Silesian University of Technology, Roosevelta 40 Street, 41-800 Zabrze, Poland; 5Key Laboratory of Advanced Carbon Materials and Wearable Energy Technologies of Jiangsu Province, Soochow Institute for Energy and Materials Innovations, College of Energy, Soochow University, Suzhou 215006, China; yangrz@suda.edu.cn; 6Leibniz Institute for Solid State and Materials Research Dresden, P.O. Box 270116, D-01171 Dresden, Germany; 7Institute of Environmental Technology, Centre for Energy and Environmental Technologies, VSB—Technical University of Ostrava, 17. Listopadu 15, 708 33 Ostrava, Czech Republic

**Keywords:** 3D graphitic shells, multi-walled carbon nanotubes, conductive polymer composite, piezoresistive sensors

## Abstract

The objective of this research was to develop highly effective conductive polymer composite (CPC) materials for flexible piezoresistive sensors, utilizing hollow three-dimensional graphitic shells as a highly conductive particulate component. Polystyrene (PS), a cost-effective and robust polymer widely used in various applications such as household appliances, electronics, automotive parts, packaging, and thermal insulation materials, was chosen as the polymer matrix. The hollow spherical three-dimensional graphitic shells (GS) were synthesized through chemical vapor deposition (CVD) with magnesium oxide (MgO) nanoparticles serving as a support, which was removed post-synthesis and employed as the conductive filler. Commercial multi-walled carbon nanotubes (CNTs) were used as a reference one-dimensional graphene material. The main focus of this study was to investigate the impact of the GS on the piezoresistive response of carbon/polymer composite thin films. The distribution and arrangement of GS and CNTs in the polymer matrix were analyzed using techniques such as X-ray diffraction and scanning electron microscopy, while the electrical, thermal, and mechanical properties of the composites were also evaluated. The results revealed that the PS composite films filled with GS exhibited a more pronounced piezoresistive response as compared to the CNT-based composites, despite their lower mechanical and thermal performance.

## 1. Introduction

Recently, significant efforts have been devoted to the development of improved flexible, tensile, and highly sensitive piezoresistive strain gauges that exhibit changes in electrical resistance in response to mechanical deformations [1]. Piezoresistive sensors, renowned for their exceptional sensitivity and ability to operate at low voltage, hold great potential for monitoring load-bearing elements in industries such as construction, automotive, aviation, and aerospace, crucial for ensuring their long-term performance [2]. Furthermore, the escalating scientific interest in piezoresistive sensors in recent years stems from their extensive applications in second-generation robotics, biology, biomedicine, rehabilitation, personal health monitoring, and wearable electronic devices such as robotic or artificial e-skin. These sensors can be either surface-mounted on elements or embedded in complex electronic devices, without significantly altering the properties of the final composite element, especially its mechanical characteristics [3].

The development of the first generation of strain gauges was closely linked to the discovery of the piezoresistive effect in semiconductor materials based on energy band theory. However, recent research has focused on a novel class of conductive polymer composites (CPCs) composed of particulate components with excellent electrical conductivity dispersed within an insulating polymer matrix. Silver nanoparticles [1,4] and carbon nanomaterials with various structures [5,6], such as graphene oxide, reduced graphene oxide, carbon nanotubes, and carbon black [7,8,9,10,11] are among the commonly employed conductive materials in piezoresistive sensors. These materials exhibit high electrical and mechanical properties and can be effectively distributed within the polymer matrix, making them promising candidates for piezoresistive strain gauges. Notably, 3D graphene materials, characterized by their distinctive shape, low density, customizable void volume, high surface area, excellent flow characteristics, and high thermal and electrical conductivity, have attracted considerable research and industrial interest [11].

Carbon nanomaterials, including 3D graphene materials, demonstrate not only good chemical and mechanical stability but also significant potential in various applications, such as catalysts, electrodes, batteries, adsorbents, materials for gas storage, templates for nanostructure manufacturing, and components of sensors. Polymer systems incorporating nanocarbon fillers offer several advantages, including relatively low cost, lightweight, ease of processing into complex-shaped products, resistance to corrosion and external factors, and controllable conductivity [12,13]. Typically, in the preparation of CPCs, conductive filler particles are randomly dispersed in the polymer matrix to establish effective conductive paths [14]. The electrical conductivity of the resulting CPCs depends on the nature, structure, shape, and dispersion state of the conductive nanocarbon particles [14]. The quality of graphene nanoparticle dispersion plays a crucial role in improving electrical parameters and is directly linked to synthesis and processing techniques, as reported in the literature [15,16]. Three primary approaches have been employed to fabricate polymer nanocomposites with graphene materials: Solution blending, in situ polymerization, and melt processing. A well-dispersed nano-additive in the matrix should exhibit adequate stability in the resulting nanocomposites [17]. However, to achieve high conductivity values, CPCs with a random filler distribution require a relatively high concentration of the conductive phase, which complicates processing and compromises the mechanical strength of the composites, making them more brittle. This approach is not always economically viable. Another viable approach to enhancing the conductivity of CPCs involves creating a segregated system of conductive carbon filler within the polymer phase, as described by Mamunya et al. [18,19,20,21]. The application of this model during composite fabrication significantly reduces the required filler content to reach the percolation threshold [22,23,24]. However, this approach does not enable the production of thin films with high homogeneity.

Polystyrene has been studied for flexible sensors, typically as an additive rather than the main polymer component. Hu et al. [25] described a process for fabricating polystyrene microspheres on graphene nanosheets (GNS) through in situ emulsion polymerization. The addition of 2 wt.% GNS led to a significant increase in conductivity compared to neat polystyrene. Self-assembled polystyrene microsphere films were applied as the microstructure layer in flexible pressure sensors with polydimethylsiloxane (PDMS) as the substrate and single-walled carbon nanotube films as the electrode material [26]. The resulting pressure sensor, with a sandwich structure, exhibited a wide pressure detection range (from 4 kPa to 270 kPa), a sensitivity of 2.49 kPa^−1^, and a response time in the tens of milliseconds range. Another study demonstrated a highly responsive, flexible piezoresistive strain sensor using laser-thermally reduced graphene oxide doped with polystyrene nanoparticles [27]. The nanoparticles altered the morphology of the sensing film by separating stacked graphene fragments and creating partially connected conducting channels, significantly enhancing the resistance change under strain. Two distinct resistance-changing mechanisms were observed with the doping of nanoparticles of different sizes. Compared to strain sensors based on graphene oxide (GO), the gauge factor of the sensor doped with 90 nm nanoparticles could reach up to 250 under small deformations (within the linear region below 1.05%).

In this study, polystyrene matrix composites with 3D graphitic hollow shells were fabricated using the solvent casting method and characterized in terms of their structural parameters, morphology, and thermomechanical properties. The influence of the graphitic shell structure on the piezoresistive response of carbon/polymer composite thin films was thoroughly discussed, with a comparison to data obtained for polystyrene systems filled with commercial multi-walled carbon nanotubes.

## 2. Materials and Methods

A commercial magnesium oxide powder with particle sizes in the range of 10–150 nm was purchased from American Elements. Granulated polystyrene (Synthos PS GP 154, Synthos, Oświęcim, Poland) with a melt flow index (MFI) of about 9 g/10 min, a Vicat softening temperature of 86 °C, and shrinkage in the range of 0.2 to 0.5% was used to prepare composites. Commercial multi-walled CNTs (C ≥ 95%, average length 5 mm, average diameter 6–9 nm) were purchased from Sigma Aldrich (St. Louis, MO, USA). Hydrochloric acid (35.38% pure p.a.), chloroform (98.5% pure p.a.), ethanol (99.8% pure p.a.), and methanol (99.8% pure p.a.) were purchased from Chempur (Piekary Śląskie, Poland).

### 2.1. Synthesis of 3D Graphitic Shells

The synthesis of nanosized, three-dimensional graphitic shells involved the use of the chemical vapor deposition (CVD) method, utilizing commercial MgO NPs as a template with precisely defined dimensions [28,29,30,31]. The growth of graphene using CVD was achieved by subjecting MgO powder to a high-temperature argon-ethyl alcohol mixture flow, with ethyl alcohol serving as the carbon source. To carry out the process, a measured amount of MgO was placed in a ceramic boat (crucible) and positioned inside a tube furnace. Next, argon flow was introduced at a rate of 10 L/min, passing initially through a container of liquid ethanol before being directed into the reaction furnace. The furnace temperature was gradually increased to 800 °C, and the sample was treated in an argon-ethyl alcohol atmosphere for 1 h. Subsequently, the furnace was cooled to room temperature with the assistance of a gas stream. The formation of MgO covered by graphene structures was visually confirmed by the presence of deep black powder material.

The synthesized MgO/C particles were then subjected to thorough washing with a diluted hydrochloric acid solution mixed with distilled water in a ratio of 1:3, continuing the washing process until the complete removal of MgO. The resulting 3D graphitic spheres, which precipitated in the solution, were further washed with distilled water and dried at 90 °C for 12 h to ensure complete water removal. The yield of the obtained graphene material, in the form of hollow carbon shells, was approximately 7.5% of the mass of the synthesized MgO/C structures.

### 2.2. Composite Preparation Procedure

The fabrication process for the low-filled nanocarbon/polymer composite materials involved the use of polystyrene as the polymer matrix. Two series of composites with different concentrations of graphene sheets (GS) and carbon nanotubes (CNTs) as nanocarbon fillers, ranging from 0.25% to 2% by weight, were prepared. The solution casting method was employed for composite preparation. To disperse GS or CNTs, ultrasonic processing in chloroform was used. Then the polystyrene granules were dissolved in mixtures of graphene nanomaterials in chloroform. Mechanical stirring was applied to ensure uniform dispersion, and subsequently, the mixtures were cast onto glass plates as solutions. The samples were left in the open air at room temperature for approximately 24 h to allow for complete solvent evaporation and the separation of the polymer composite film from the glass surface.

### 2.3. Characterization and Instruments

Scanning electron microscopy (SEM) studies were conducted using the SEM FEI Quanta 250 FEG (FEI Company, Hillsboro, OR, USA), equipped with a secondary electron detector (EDT), employing both high-vacuum and low-vacuum secondary electron techniques. The accelerating voltages used ranged from 10.0 to 20.0 kV. To fix the powdered samples of GS and CNTs, carbon tape was utilized on a measuring holder. Quantitative elemental analyses of the carbon materials’ surfaces were performed using SEM in conjunction with energy-dispersive X-ray spectroscopy (EDS). The fracture surface of graphene/PS composites was examined.

Transmission electron microscopy (TEM) investigations were carried out using a Tecnai G2 F20 microscope (FEI Company, Hillsboro, OR, USA), operating at an acceleration voltage of 200 kV. TEM images were recorded using the Gatan Rio 16 CMOS camera (Gatan, Pleasanton, CA, USA) and processed with Digital Micrograph software (Gatan 2.x). For the TEM studies of graphene materials, ethanol was used to disperse the samples through ultrasonic treatment. The resulting suspension was placed on a 200-mesh copper grid (Quantifoil, Großlöbichau, Germany), and after alcohol evaporation, the specimens were analyzed within the microscope chamber.

XRD measurements were performed using a D8 Advance diffractometer (Bruker AXS, Karlsruhe, Germany) with a Cu-Kα cathode *(λ* = 1.54 Å). The scan rate was 0.6°/min with a scanning step of 0.02° in the range of 5° to 90° 2Θ, using Bragg–Brentano geometry. The fitted phases were identified using the DIFFRAC.EVA program with the ICDD PDF#2 database. Lattice parameters, crystal size, and lattice strain were calculated using Rietveld refinement in the TOPAS 6 program, based on the Williamson–Hall theory [32,33,34]. The pseudo-Voigt function was used in the description of diffraction line profiles at the Rietveld refinement. The R_wp_ (weighted-pattern factor) and S (goodness-of-fit) parameters were used as numerical criteria for the quality of the fit calculated from experimental diffraction data [35].

The particle size distribution of MgO particles was determined using a Zetasizer NANO ZS (Malvern, Malvern, UK) employing the technique of dynamic light scattering (DLS) with photon correlation spectroscopy (FCS), employing non-invasive backscattering (NIBS) technology. To conduct these studies, 0.1 wt.% suspensions of MgO were prepared in methanol and sonicated for 10 min at maximum power.

Raman spectra were recorded using a Witec Alfa M300+ spectrometer with an Nd-YAG laser beam operating at an excitation wavelength of 532 nm and a laser power of 50 mW. The measurement parameters were as follows: Laser power—1 mW, time of exposure—3 s, number of scans—100, and an acquisition range of 0–3800 cm^−1^. Approximately 5 points were analyzed for each sample to gain insight into sample structural homogeneity using the spectrometer in live mode, without recording the spectra. The ratios of the intensity of the D and G peaks (ID/IG) and their areas (AD/AG), which estimate the ordering of the graphene sheets of GS and CNTs, were determined using Witec Project 4.1 software.

Calorimetric measurements were performed using a differential scanning calorimeter (TA Instruments DSC 2000) in a dry nitrogen atmosphere with a nitrogen flow of 50 mL/min. Non-hermetical aluminum pans were used to hold approximately 10 mg of samples. The first dynamic measurements were carried out from 0 to about 30 °C above the initial glass transition temperature of the polymer matrix to standardize all composite samples. The correct glass transition temperature (T_g_) of PS and its composites with GS and CNTs was determined during the second scanning dynamic runs performed directly after the first ones, from 0 to approximately 160 °C with a heating rate of 20 °C/min. The inflection point on the heat flow vs. temperature curve was identified as T_g_.

Dynamic mechanical analysis (DMA) was conducted using a DMA 2980 TA Instruments analyzer. Thin film samples with thicknesses of 0.1–0.2 mm, widths of 5–7 mm, and lengths of 6–8 mm were analyzed using a film tension clamp. The samples oscillated at a single frequency of 1 Hz with an oscillation amplitude of 5 mm. A static force of 0.01 N and an auto strain of 125% were applied to the sample. Mechanical spectra were obtained with a temperature trace rate of 3 °C/min, ranging from 0 °C to above 200 °C. Changes in the storage modulus (E’), loss modulus (E”), and loss factor (tan δ) were recorded as a function of temperature. The temperature at which the maximum of the E” peak was observed was considered the glass transition temperature (T_g_) of the tested samples.

The mechanical properties of the GS/PS and CNTs/PS composite films were tested using a Materials Testing Machine 4204 (Instron, Norwood, MA, USA) at room temperature, with a tensile deformation rate of 100 mm/min. Samples with a length of 50 mm and a width of 5 mm were used for the tests, and the sample thickness was measured with a micrometer. Each sample was tested five times under the same conditions.

The electrical properties of the composites at room temperature, with a direct current applied to the samples, were determined using rectangular samples with dimensions of 20 × 10 × 0.2 mm. High-purity silver electrodes were pasted on both sides of the sample. The distance between electrodes was set at 10 mm. The resistance of the samples was then measured using a UT804 True RMS Multimeter (UNI-T, Dongguan, China). Conductivity was calculated using Equation (1).
(1)$$\sigma =\frac{L}{R\cdot S}$$
where δ is conductivity (S/cm), L is the distance between electrodes in cm, R is sample resistance in Ω, and S—cross-section in cm^2^.

The piezoresistive response of the composites was measured under tensile and bending stress using a Keithley 6485 picoamperometer connected to LabVIEW 13.0 software. An initial voltage of 10 V was applied using an external power source, and the sample was left under this voltage for 300 s to obtain stable conductivity values. Sample dimensions for the tensile tests were 65 × 5 × 0.2 mm, while for the three-point bending tests, dimensions of 20 × 10 × 0.2 mm were used. Silver paste was applied on both sides of the measuring length of the sample, resulting in a distance of 0.8 cm between electrodes, with a width of 1 cm. The sensors were isolated with insulating tape and fixed in a Deben Microtest MT2000 tensile and compression stage, which was used to measure the tensile strain. Tensile and bending tests were performed using a DEBEN microtensile setup with a 2 kN crosshead and DEBEN 6.3.40 software. The crosshead speed during tensile tests was set to 0.4 mm/min, and the working distance between clamps was set to 15 mm. For bending tests, the distance between clamps was also 15 mm, and bending was performed using an external tip. Piezoresistive responses were measured both in the initial and deformed states, with a step Δ = 0.2 mm, using a multimeter UNI-T UT804 (Uni-Trend Technology, Dongguan City, China). The relative position (y) of the device was measured by a displacement sensor (Mitutoyo Absolute Digimatic Indicator ID-C 12.7 mm, Mitutoyo Poland, Wrocław, Poland) with an accuracy of 1 µm.

## 3. Results

### 3.1. Characterization of Graphitic Shells

The synthesis of nanosized 3D graphene was carried out by the CVD method using commercial magnesium oxide nanoparticles as a template. It is well known that the CVD method makes it possible to obtain macroscopically homogeneous 3D graphene with improved structural stability due to the absence of defects as well as high electrical conductivity [30,31]. The structure (Figure 1a,b) and size distribution of the initial MgO particles (Figure 1c) were studied by TEM microscopy, which made it possible to estimate the possible sizes of the synthesized graphene material.

It is shown that the original magnesium oxide is a rather heterogeneous sample with an average particle size of about 30–70 nm. The particle size distribution estimated using TEM images correlates well with the data obtained for 0.1 wt.% suspensions of MgO particles in methyl alcohol using the DLS method (Figure 1d). The analyses performed confirm the formation of three-dimensional graphene structures on the matrix of magnesium oxide particles. Thus, SEM-EDS and TEM analysis showed differences in the morphology of the material at each stage of production, which is confirmed by the deposition of graphene layers on magnesium oxide particles (Figure 2). Qualitative energy dispersive analysis (Figure 3b) and phase composition analysis by X-ray diffraction (Figure 4) confirmed the presence of carbon.

The SEM (Figure 3a) and TEM (Figure 3c,d) photographs of the final graphitic shells after the removal of MgO confirmed obtaining a hollow graphitic sphere. The EDS analysis of the obtained graphitic shells (Figure 3b) showed that the carbon content in atomic percent was 95.7 and oxygen was 4.3. The presence of magnesium after the dissolution of the MgO matrix was not observed.

XRD patterns of initial MgO, MgO covered with graphene, and graphitic shells after removal of MgO are presented in Figure 4. A highly crystalline structure of initial MgO powder, corresponding to cubic magnesium oxide (Fm-3m crystal group) with lattice parameter a = 4.211 Å, has been confirmed. The calculated crystallite size is about 51 ± 3 nm, which remains in good agreement with TEM observations and DLS results. After covering MgO nanoparticles with graphene sheets during the CVD process, a slight halo around 25° 2Θ was found, suggesting the presence of amorphous carbon on the surface of MgO particles. It should be noted that the layer of graphene on the surface of MgO particles, as observed in TEM images (Figure 3c,d), is below 5 nm. Therefore, its presence on the XRD pattern of MgO/GS in a well-pronounced manner is rather unlikely. The XRD pattern of hollow carbon shells after etching of MgO exhibits a typical amorphous nature, suggesting no ordering of carbon layers during their growth. It should be noted that the organization of carbon layers into a crystal structure requires a very low misfit between single layers. TEM pictures (Figure 3c,d) confirm a rather random distribution of single carbon layers formed during the growth of shells, which explains the lack of their crystal structure.

Raman spectroscopy, an ideal tool, was used to confirm the formation of a few layers of graphene material. This is possible because such graphene in Raman spectroscopy gives two distinct bands due to the first-order peaks around 1350 and 1600 cm^−1^, which are called the D and G bands, respectively. The initial MgO powder was characterized by the prominent Raman bands at 280, 472, 2438, and 2885 cm^−1^ [36]. The band around 472 cm^−1^ is assigned to E2 (high) first-order Raman modes. The band around 2880 cm^−1^ corresponds to the C–H stretching mode. The Raman spectrum of the MgO sample after the CVD process showed the appearance of the D and G bands, which confirms the production of a graphene-layered material (Figure 5). In the Raman spectra of the formed graphitic shells, the G band characteristic of graphite-like materials with a maximum at 1585 cm^−1^, the D band related to defects in carbon nanotubes at 1336 cm^−1^, the G’ band at 2670 cm^−1^ corresponding to the first overtone of the D-mode, and the band D’ at 2870 cm^−1^ corresponding to the first overtone of the D-mode, were observed.

The ratio of the intensity of the D and G peaks (I_D_/I_G_) in the Raman spectra used to conclude the degree of defectiveness of the obtained material was about 1.3. These values make it possible to judge structural defects in graphene materials since a higher ratio suggests structural defects relevant to sp^3^ hybridization associated not only with some irregularities in the graphene layers, such as pores, impurities, or other symmetry-breaking defects but also with a relatively large number of dangling edges, which leads to sp^3^ hybridization of carbon [30].

### 3.2. Characterization of PS Nanocomposites

Low-filled nanocarbon/polymer composite materials were fabricated using polystyrene as the polymer matrix. The influence of graphene materials’ structure (GS, CNTs) and their concentration on the morphology and properties of the synthesized graphene/polymer nanocomposites was determined. Scanning electron microscopy (SEM) was employed to characterize the morphology of the fracture surface of pure polystyrene and its composite films filled with the lowest (0.25 wt.%) and highest (2 wt.%) content of GS and CNTs, as well as the distribution of graphene fillers within the polymer matrix (Figure 6).

The fracture surface of pure PS (Figure 6a) appears flat, without ridges or concavities, displaying signs of enhanced matrix ductility. The fracture surface character remains consistent from the notch to the end of the fracture. SEM images of the cross-section of GS/PS films clearly reveal the distribution of graphene materials within the composite, indicating strong physical interactions between the graphitic shells and the polystyrene phase (Figure 6b,c). However, at a concentration of 0.25 wt.% graphitic shells, uniform filler distribution in the PS matrix and the formation of a conductive path of GS NPs were not achieved, unlike carbon nanotubes, which formed effective connections in the matrix at the same concentration. Aggregation of 3D graphitic shells was also observed, likely resulting from their sedimentation in the lower part of the composite film. At a graphene content of approximately 2 wt.%, the filler practically fills the entire PS matrix, forming a continuous network of graphene particles within the polymer matrices. This, in turn, enhances electrical conductivity and leads to more stable sensing responses when used as sensors. SEM analysis confirmed the presence of defects in polystyrene composite films synthesized via the casting solution method, likely associated with the removal of solvent vapors during film drying. However, as the filler concentration increased, these defects became smaller, and at a graphene content of around 2 wt.%, they almost disappeared, potentially improving the mechanical characteristics of PS composites. The thickness of the PS-based films, measured using SEM, was approximately 200 µm, and the obtained data were utilized to calculate the resistivity of the resulting materials.

The DSC results elucidate the influence of graphitic shells and carbon nanotubes on the thermal behavior of PS-based composites. Table 1 provides a summary of the glass transition temperature (T_g_) for the unfilled polymer film and PS composites. DSC analysis showed that in the presence of graphene structures, both 3D graphitic shells and carbon nanotubes, in the PS matrix, T_g_ increased by approximately 10–12 °C for all composites compared to the pure PS matrix. This increase can be attributed to the higher restrictions on chain movement within the polymer phase, possibly resulting from the entrapment of intercalated polymer chains within the graphene layers, which hinders segmental movement. It is speculated that conducting the first DSC runs above the T_g_ of polystyrene contributes to an increased degree of intercalation of graphene layers by PS chains.

It is well known that graphene nanomaterials can significantly impact the mechanical properties of polymer matrix composites. DMA analysis enables the observation of physical and chemical changes in the polymer phase and filler–polymer interfaces, shedding light on the dynamic mechanical properties of graphene–polymer composites. The changes in storage modulus and loss factor values under sinusoidal external loads over a wide temperature range are valuable for characterizing the distribution of filler nanoparticles within the polymer phase and at the filler–polymer interfaces. Table 1 provides a comprehensive summary of the viscoelastic parameters in the glassy, rubbery, and glass transition regions for unfilled polystyrene matrix as well as PS-based composites with varying contents of graphitic shells and carbon nanotubes. Figure 7 illustrates the temperature dependencies of storage modulus, loss modulus, and loss factor for pure polystyrene and its composites with different contents of graphitic shells and carbon nanotubes.

The storage modulus reflects the material’s stiffness and its ability to return to its initial state after the applied force is removed. Analysis of the storage modulus values in the glassy state for PS composites filled with CNTs reveals that, compared to the pure PS matrix, the highest increase in storage modulus is observed for composites with lower CNT content, confirming the reinforcing effect of this nanofiller. The E_’g_ value for the 0.25%CNT/PS composite is approximately 18% higher than that of pure polystyrene. In contrast to CNTs, when spherical GSs are used as nanofillers in polystyrene, there is no observed enhancement in the dynamic mechanical properties; instead, noticeable decreases in storage modulus values for GS/PS composites are observed compared to pure PS and CNT-filled composites. This is likely due to the local agglomeration of GS nanoparticles within the PS phase, as confirmed by SEM observations. The static mechanical tests exhibit a similar trend (Figure 8a, Table 2), with the exception of the PS composite with 1%GS, which shows an increase in modulus compared to pure unfilled polystyrene.

Polystyrene, being a conventional thermoplastic material, exhibits a low storage modulus in the post-glass transition state. In the case of PS composites, whether with graphitic shells or carbon nanotubes, the values of E’_g_ and E’_r_ actually decrease, contrary to the usual expectation of an increase in modulus with the addition of a nanofiller in the highly elastic or plastic state. Only the 2%GS/PS composite showed a higher storage modulus after the glass transition region. Considering the very high storage modulus of pure graphene layers (~1 TPa), one would anticipate significant increments in the E’_g_ and E’_r_ moduli for polymer composites with graphene structures. Therefore, the lower E’_r_ values observed for PS composites could be attributed to the higher tendency of graphene filler particles to agglomerate, their inhomogeneous dispersion, or poorer adhesion to the polymer matrix, as confirmed by SEM results. Literature reports on polymer nanocomposites with varying content of graphene materials often exhibit divergent results, showing both significant improvements and variations [37,38] or an imperceptible increase in the E’ value, especially in the glassy state of polymers [39,40].

The loss modulus curves obtained from the DMA experiments are presented in Figure 7b,e. In terms of the loss modulus value at peak temperature, which characterizes the ability of a viscoelastic material to dissipate mechanical energy, a clear reinforcing effect was observed in CNTs/PS composites, with the largest increase of approximately 67% in the case of the 1%CNT/PS composite compared to the pure PS matrix. The higher damping observed in CNT/PS composites can be attributed to the uniform dispersion of nanotubes in the polymer matrix. The increase in loss modulus for GS/PS composites was also observed for all composites except for polystyrene with 0.25 wt.% of graphitic shells. The maximum temperature of the loss modulus, similar to the inflection temperature of E”, marks the glass transition temperature of the graphene/PS composites. The glass transition temperature (T_g_) can be measured using various techniques such as DSC, DMA, TMA, rheology, and dielectric spectroscopy. DMA provides a large possibility to determine T_g_ using the temperature dependence of storage modulus, loss modulus, and tan δ, which differ in values compared to other methods, including DSC. The T_g_ values of the nanocomposites evaluated from the E’ and E” curves are listed in Table 1. The T_g_ values obtained from the loss modulus did not show notable changes, contrary to the DSC results. A small increase in T_E”_ was observed for all CNT/PS composites. In the case of GS/PS composites, a reduction in T_g_ was observed for the composite with 0.25% graphitic shells, while for the other composites, the T_g_ values practically remained unchanged compared to the PS matrix. However, different T_g_ values were found when considering the inflection temperature of the storage modulus drop in the glass transition region of polystyrene (Table 1). Both GS/PS and CNT/PS composites exhibited an increase in these temperature values compared to PS. The exception was the 0.25% GS/PS composite, which showed a reduction of 2.5 °C in T_g_ compared to the PS matrix and a decrease in the range of 10–15 °C compared to other composites.

For all PS composites, both with GSs and CNTs, a reduction in tan δ values was observed, with the greatest decrease observed in the 1%CNT/PS and 2% GSs/PS composites, while the smallest reduction was found in composites with the lowest content of GSs and CNTs. At the same time, an increase in the half-height width of the tan δ peak was observed for all composites. This could be attributed to the interactions between polystyrene and the high surface area of graphene fillers, which restrict the molecular motions of polystyrene chains, resulting in an extension of the PS glass transition temperature range. This phenomenon has been explained by Lu and Nutt in their study on organically modified layered silicate–epoxy nanocomposites [41]. From this perspective, the molecular relaxation of polystyrene during the glass transition can be divided into two regions: One with slowly relaxing domains tethered to the graphene layers and another with faster motion of the bulk polymer material, having the same glass transition as polystyrene without graphene inclusions. The presence of a higher content of slowly relaxing polystyrene areas occurs when graphene fillers are finely dispersed in the polymer matrix, resulting in a significant increase in the glass transition temperature. Therefore, the T_g_ and S_1/2_ (tan δ) values can be used as useful indicators of the level of graphene nanofiller dispersion in polymer composites.

Since the filled polymer films are intended for use as strain sensors, mechanical property tests were conducted to evaluate their strength. Figure 8 illustrates Young’s modulus, elongation at break, and tensile strength for PS/3D graphene and CNT/PS composites.

The mechanical properties of polymer composites with graphene fillers, as expected, depend on the structure and concentration of the filler in the polymer matrix. The stress–strain curves for GS/PS and CNT/PS samples illustrate the relationship between mechanical properties and the degree of polymer filling with carbon materials (Figure 8). Detailed results of the mechanical evaluation, including Young’s modulus, maximum stress, and elongation at break, were determined and calculated based on the stress–strain curves and are presented in Table 2.

It can be observed that an increase in the number of carbon nanostructures leads to a higher Young’s modulus in composites with CNTs. In composites filled with graphitic shells only, the composite with 2 wt.% of filler exhibits a higher Young’s modulus value. For composites with lower GS content (0.25 and 0.5 wt.%), the tensile strength values were improved compared to the pure PS matrix, and the elongation at break values were comparable. However, for composites filled with higher amounts of graphitic shells (1 and 2 wt.%), a noticeable reduction in these parameters was observed. In particular, an increase in Young’s modulus, accompanied by three times lower elongation than that of pure PS and low-filled GS/PS composites, was detected in the composite with 2 wt.% GS. This behavior can be explained as a result of particulate-like reinforcement of the polymer matrix in the case of spherical GS.

The influence of graphene material’s structure, concentration, and organization on the electrical characteristics of the fabricated composites based on polystyrene was studied using DC conductivity measurements. As expected, composites containing CNTs exhibited the best electrical properties (Figure 9), which can be attributed to the intrinsically high electrical conductivity of the nanotubes. Furthermore, all fabricated samples with CNTs demonstrated conductivity characteristics typical of semiconductors. It was also observed that composites with GS filler displayed lower conductivity compared to those with CNTs, reflecting the differences in the morphology of the carbon nanomaterials. However, for both series of composites, the conductivity increased with an increasing amount of nanocarbon.

It was demonstrated that the electrical conductivity of polymer films filled with nanotubes decreased linearly with deformation (Figure 10b,d), whereas for composites filled with 3D graphene, this dependence exhibited a more complex pattern (Figure 10a).

The conducted studies confirm that GS/PS composites exhibit a different type of piezoresistive response compared to composites with CNTs. The inclusion of hollow carbon shells has a significant impact on the sensor’s response during the bending of the composite, which is attributable to the deformation of the spherical graphitic structure. This deformation leads to the absence of a response under elastic deformation, but when the strain exceeds the elastic limit, the sensitivity of 3D graphene/polymer composites becomes remarkably high, resulting in an extremely large change in sensor resistance.

Figure 10 presents the responses of GS/PS and CNT/PS composites with 0.5% carbon nanostructures under bending and tensile stress, while cycling tests are shown in Appendix A (see Appendix A). Different behaviors can be observed in both composites under bending and tensile stress, with detectable differences between the carbon nanomaterials. In bending tests, the GS-reinforced composite sensor recovers almost to the same level after deformation, while the one containing CNTs experiences a slight decrease in conductivity with each step until failure. Furthermore, the response of the PS/GS composite undergoes a significant change in behavior beyond a deflection of 1.5 mm, leading to a substantial conductivity change, whereas the response of the PS/CNT composite remains relatively consistent. Similar patterns are observed in the behavior of the sensors under tensile stress, with minor differences. The PS/GS composite exhibits consistent responses with a stable relative conductivity change, while the PS/CNT composite displays a higher variation in relative conductivity changes. Additionally, the sensor with CNT addition shows a tendency towards higher recovery, which is a result of its mechanical and elastic properties, particularly its greater ability to undergo elastic deformation compared to the PS/GS composite.

The detected results stand in good agreement with literature data, especially about polymer composites with CNT addition. For example, Spinelli et al. [5] and Wang et al. [6] described epoxy-based composites with a mean change of resistance in the range of 1 to 1.5% for epoxy/0.3% MWCNT and epoxy/3% CNT/GNP mixtures, respectively. Both teams [5,6] used various carbon nanomaterials and their amounts, and the results obtained for elastic deformation are very similar. Other applications of CNT in polymer composites are also described in the literature, as in the work of Shen et al. [42]. The possibility of detecting GFRP delamination by carbon nanotube bucky papers was investigated, and results show resistance changes in the range of 40 to 2000%, with a strong dependence on the glass fiber fabric orientation. Another possibility of CNT application was described by Wang et al. [43]. The potential of CNT-coated carbon fibers in glass fabric to detect laminate deformation was investigated, with influence depending on the applied coating on carbon fiber. The resistance change varies from 0.7 to 6% at laminate break. Alamusi et al. [44] described the impact of the CNT amount in a polymer matrix on the resistance change and noticed that the best responses (about 12–14%) were obtained for samples with a lower volume fraction of carbon nanomaterials. Literature also reports the application of elastomer-based composites for breath [45], pressure [46], and strain [47] detection with responses higher than 20% of initial resistance. However, as used in the mentioned studies, polymer matrices show significantly higher elasticity than polystyrene.

The models shown in Figure 11 and Figure 12 illustrate the changes in the conductive path under bending for GS/PS (Figure 11) and CNT/PS (Figure 12) composites, revealing a fundamental difference between these two materials. In their initial states, both materials exhibit a conductive path formed by carbon material nanoparticles, facilitating the flow of direct current through the material. However, under bending deformations, the conductive path is disrupted. Due to the different morphologies of carbon fillers, this effect manifests at various intensities, leading to different changes in the conductivity of the composite. It is evident that this phenomenon is more pronounced for graphitic hollow spheres than for carbon nanotubes, due to their particulate and fibrous morphologies, respectively. The distances between nanotubes remain relatively unchanged, given their limited mobility in the polymer matrix. In contrast, the distances between graphitic hollow spheres were initially larger, making even slight changes have a greater impact on the conductivity of the composite conductivity.

## 4. Conclusions

Comparative studies were conducted to investigate the effect of spherical hollow graphitic shells and carbon nanotubes on the morphology, thermomechanical, and piezoresistive properties of polystyrene (PS)-based composites. Based on the findings, the following conclusions can be drawn:The addition of carbon nanotubes (CNTs) to PS improved the storage modulus of all CNT/PS composites, except for the composite with the highest CNT content (2 wt.%), due to the increased stiffness associated with dispersed carbon structures having a relatively high aspect ratio in the matrix. However, the use of spherical graphitic shells (GS) as a nanofiller in a PS matrix did not show an enhancing effect on the storage modulus. On the contrary, noticeable decreases in the storage modulus values of GS/PS composites were observed compared to pure PS and composites with CNTs. This decrease can be attributed to the local agglomeration of GS nanoparticles in the PS phase, as confirmed by scanning electron microscopy (SEM) observations. The results of Young’s modulus under tensile deformation for the studied composites were consistent with the dynamic mechanical analysis (DMA) results. The elongation at break values showed improvement for all composites with CNTs, except for the composite with 2 wt.% of CNTs, while a reduction in this parameter was observed for all composites with GS.SEM observations revealed that a concentration of 0.25 wt.% of graphitic shells was insufficient to achieve a uniform distribution of the filler in the PS matrix and to create a conductive path of GS nanoparticles, unlike carbon nanotubes, which formed effective connections in the matrix at a concentration of 0.25 wt.%. The application of 0.5 wt.% of both graphitic shells and carbon nanotubes resulted in composites with conductivity similar to semiconductors, which is highly desirable for piezoresistive strain sensors.Piezoresistive tests under bending and tensile stress showed that composites with graphitic shells exhibited higher stability of response compared to those with carbon nanotubes, which can be attributed to the unique morphology of graphitic shells. The use of hollow spheres allowed for good composite conductivity in the initial state and high sensitivity under deformation, which cannot be achieved using multi-walled carbon nanotubes.The polystyrene-based composites filled with graphitic hollow spheres exhibit exceptional piezoresistive characteristics during bending, making them suitable for use as an active element of flow sensors dedicated to both gaseous and liquid media. Another promising application is the detection of bending in various construction and building structures, including, for example, bridges, pillars, etc.

## Figures and Tables

**Figure 1 polymers-15-04674-f001:**
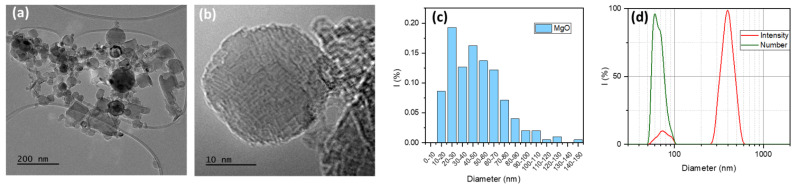
Characteristic of structure and particle size distribution of initial MgO particles by TEM (**a**–**c**) and DLS analysis for 0.1 wt.% methanol suspension of MgO NPs (**d**).

**Figure 2 polymers-15-04674-f002:**
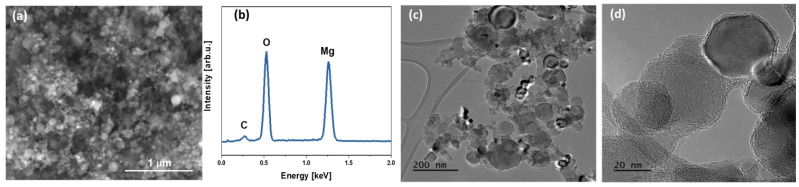
Structure of synthesized MgO/C particles: SEM image (**a**) and TEM images (**c**,**d**) and elemental composition by EDS method (**b**).

**Figure 3 polymers-15-04674-f003:**
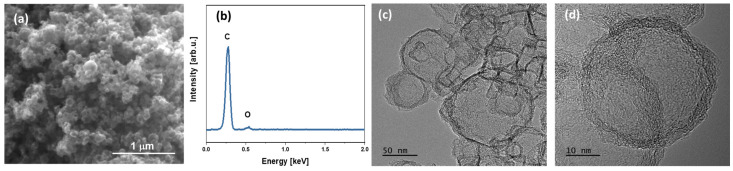
SEM (**a**) and TEM (**c**,**d**) images of 3D graphene shells obtained after MgO dissolution and EDS spectrum (**b**) of synthesized graphitic shells GS.

**Figure 4 polymers-15-04674-f004:**
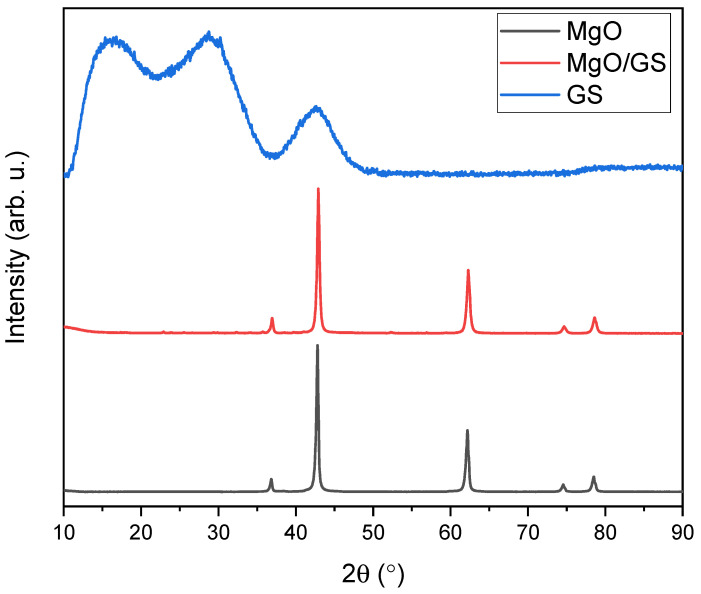
X-ray diffraction (XRD) patterns of the initial MgO nanoparticles, MgO nanoparticles covered by graphene layers, and GS after removal of MgO.

**Figure 5 polymers-15-04674-f005:**
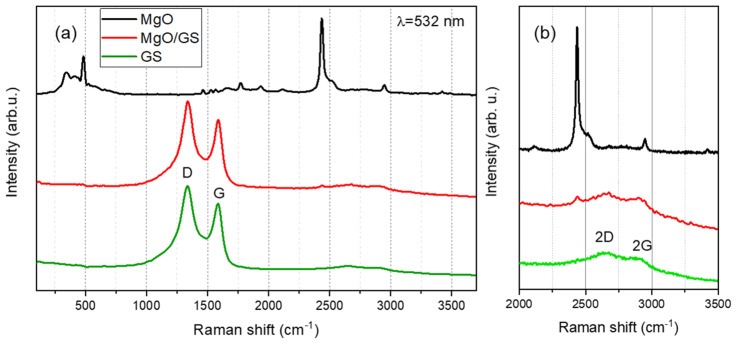
Raman spectra of MgO NPs, on the MgO template and graphitic shells (**a**) and fragment of the Raman spectrum in the range 2000–3500 cm^−1^ (**b**).

**Figure 6 polymers-15-04674-f006:**
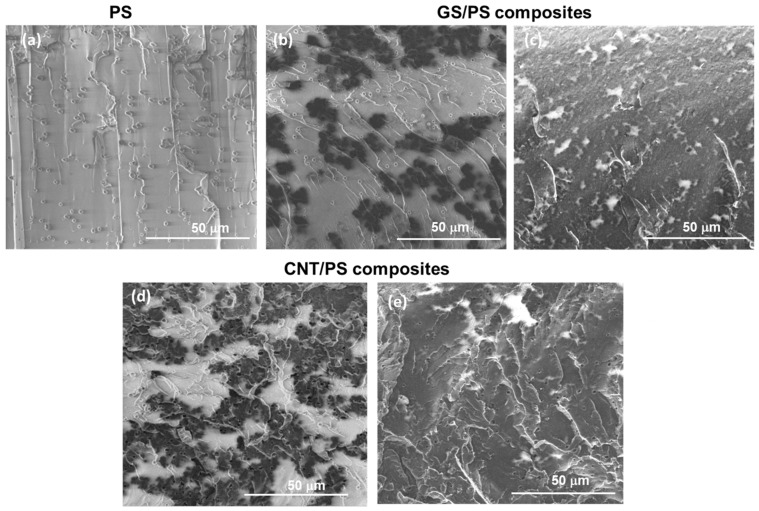
SEM images of brittle fracture surface of PS film (**a**) and GS/PS composites: (**b**) 0.25 wt.% of GS and (**c**) 2 wt.% of GS (**c**); and CNT/PS composites: (**d**) 0.25 wt.% of CNTs and (**e**) 2 wt.% of CNTs.

**Figure 7 polymers-15-04674-f007:**
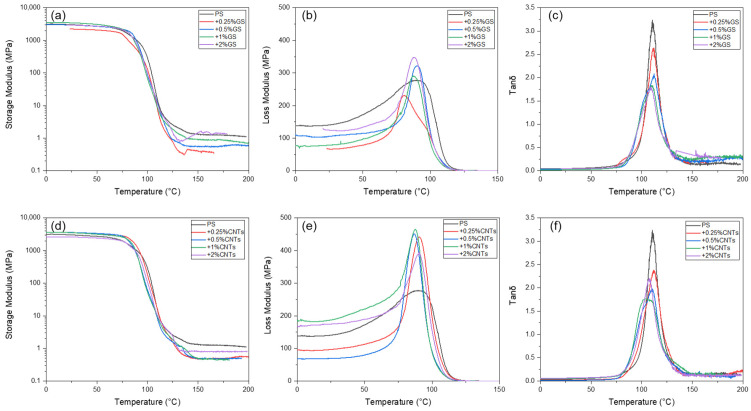
Temperature dependences of viscoelastic properties vs. temperature for GS/PS composites (**a**–**c**), and for CNT/PS composites (**d**–**f**).

**Figure 8 polymers-15-04674-f008:**
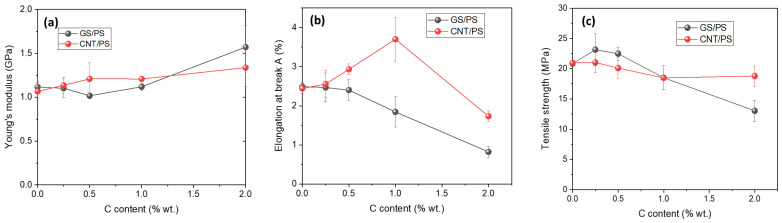
Dependence of the Young modulus (**a**), elongation at break (**b**) and tensile strength (**c**) of PS-based composite films on the content of graphene shells and carbon nanotubes.

**Figure 9 polymers-15-04674-f009:**
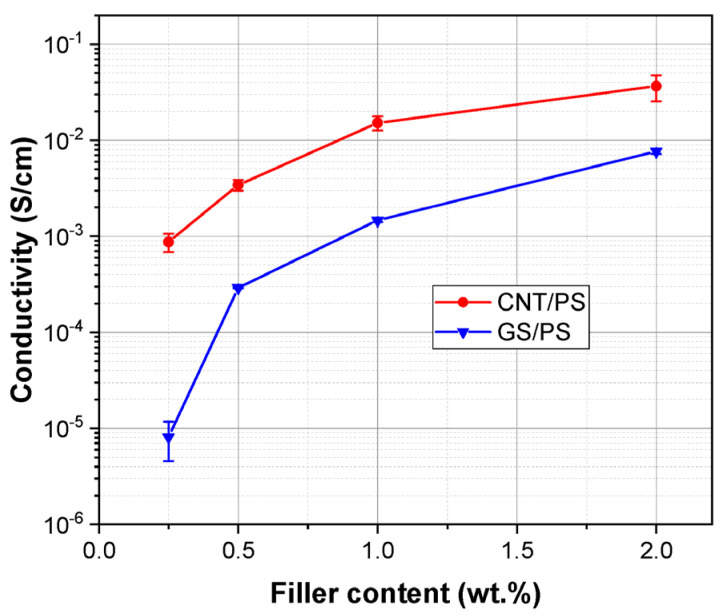
Conductivity as a function of the graphene material content in PS composites.

**Figure 10 polymers-15-04674-f010:**
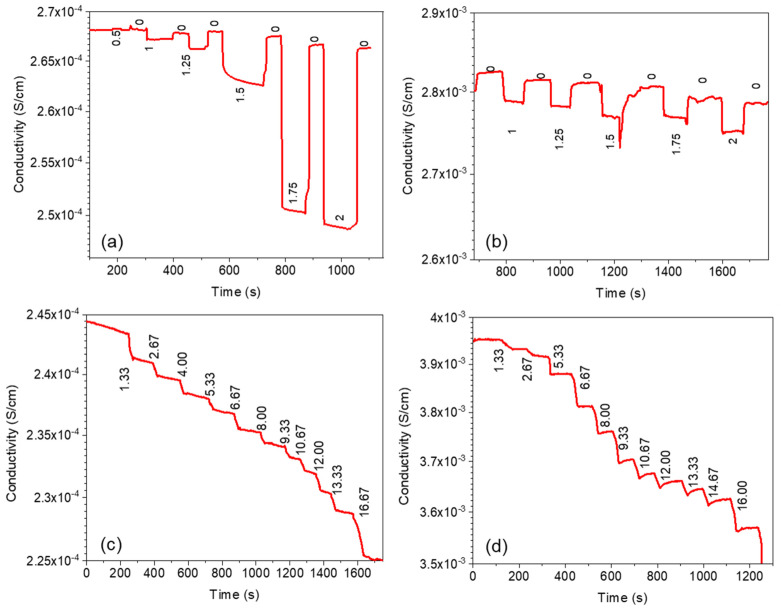
Change of conductivity as a function of time during bending tests for increased deflection (**a**,**b**) and during tensile tests for increased strain (**c**,**d**) for GS/PS (**a**,**c**), CNT/PS (**b**,**d**). Numbers on (**a**,**b**) correspond to applied deflection (mm), while on (**c**,**d**) to applied strain (%).

**Figure 11 polymers-15-04674-f011:**
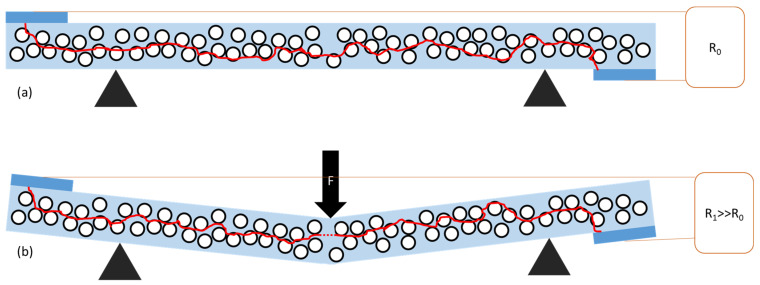
Model of change in the conductive path in GS/PS composite in the initial state (**a**) and under bending (**b**). Light blue color represents the polymer matrix, dark blue—electrodes, black circles with white interior represent graphitic hollow shells, while red color represents a schematic path of current flow through the material. Red dotted line (**b**) marks disruption of conductive path.

**Figure 12 polymers-15-04674-f012:**
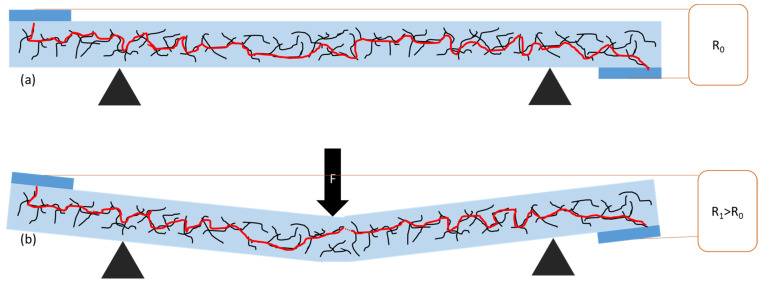
Model of change in the conductive path in CNT/PS composite in the initial state (**a**) and under bending (**b**). Light blue color represents polymer matrix, dark blue—electrodes, black lines represent multi-walled carbon nanotubes, while red color represents a schematic path of current flow through the material. Red dotted line (**b**) marks disruption of the conductive path.

**Table 1 polymers-15-04674-t001:** DSC results and dynamic mechanical parameters for PS and its composites with GS and CNTs.

Polymer System	DSC	DMA						
	T_g_ [°C]	*E*’_g_ [MPa]	*E*’_r_ [MPa]	*T_infE’_* [°C]	*E*”_max_ [MPa]	*T*_E”_ [°C]	tan *δ*	*S*_1/2_ (tan δ)
PS	81.4	3082	1.258	78.9	278	87.1	3.152	13.2
+GS								
0.25%GS/PS	92.6	2229	0.385	76.9	231	80.9	2.601	14.1
0.5%GS/PS	91.1	2980	0.541	87.1	322	90.1	2.036	23.2
1%GS/PS	92.4	3487	0.489	86.5	291	87.6	1.812	21.0
2%GS/PS	93.7	2941	1.507	86.0	347	87.9	1.771	19.6
+CNTs								
0.25%CNTs/PS	92.6	3632	0.487	90.1	442	90.5	2.351	18.8
0.5%CNTs/PS	93.6	3628	0.477	80.8	451	86.7	1.961	22.5
1%CNTs/PS	91.4	3588	0.466	87.4	465	87.4	1.761	25.3
2%CNTs/PS	93.2	2640	0.796	92.2	394	90.5	2.177	17.2

Notation: *E*’_g_—storage modulus at glassy state (25 °C); E’_r_—storage modulus at plateau region of rubber-like state (*T*_g_ + 50 °C); T_iE’_ storage modulus curve inflection point; *E*”_max_—loss modulus at peak temperature, *T*_E”_ temperature of loss modulus peak; *tan* δ—maximum value loss factor peak.

**Table 2 polymers-15-04674-t002:** Mechanical parameters for non-filled PS and its composites with GS and CNTs.

Polymer System	Thickness [mm]	Young’s Modulus *E* [GPa]	Tensile Strength R_m_ [MPa]	Elongation at Break A [%]
PS	0.22 ± 0.03	1.12 ± 0.05	20.85 ± 0.36	2.49 ± 0.10
+GS				
0.25GS%/PS	0.26 ± 0.05	1.10 ± 0.11	23.14 ± 2.62	2.47 ± 0.37
0.5GS%/PS	0.27 ± 0.06	1.02 ± 0.03	22.47 ± 1.11	2.40 ± 0.37
1GS%/PS	0.26 ± 0.01	1.12 ± 0.04	18.50 ± 1.99	1.85 ± 0.38
2GS%/PS	0.28 ± 0.04	1.57 ± 0.25	13.04 ± 1.78	0.82 ± 0.14
+CNTs				
0.25CNTs%/PS	0.34 ± 0.07	1.14 ± 0.09	21.03 ± 1.71	2.56 ± 0.34
0.5CNTs%/PS	0.29 ± 0.02	1.21 ± 0.18	20.09 ± 1.69	2.93 ± 0.13
1CNTs%/PS	0.27 ± 0.05	1.21 ± 0.03	18.46 ± 0.03	3.70 ± 1.56
2CNTs%/PS	0.24 ± 0.02	1.34 ± 0.22	18.78 ± 1.66	1.73 ± 0.13

## Data Availability

The data presented in this study are available upon request from the corresponding author.

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
