# Peer review of "Flexible Piezoresistive Polystyrene Composite Sensors Filled with Hollow 3D Graphitic Shells"

_polymers, 2023, doi:10.3390/polym15244674_

Round 1

Reviewer 1 Report

Comments and Suggestions for Authors

This work reports a polystyrene matrix composite with 3D graphitic hollow shells for flexible piezoresistive sensor. The author characterized in terms of their structural parameters, morphology, and thermomechanical properties, which is meaningful for the development of pressure sensor. However, the manuscript still has many problems.

Here are my concerns:

1.     According to previous reports, piezoresistive sensors with good properties have been widely reported, and the performance advantages of the polystyrene matrix composite with 3D graphitic hollow shells as a piezoresistive sensor demonstrated in this work seem to be unclear.

2.     The microscopic structure and deformation mechanism of this pressure sensing material is not clear, is it possible for the author to add a model diagram for further introduction?

3.     The data demonstration of the sensing properties of the material is insufficient, for example, what is the sensitivity of the device constructed of this material, and whether it can respond stably under cyclic strain?

4.     The figures shown in Fig 10 are puzzling. Do they refer to strain percentages? It is recommended to specify this in the annotations of the figure.

5.     In the manuscript, there is little description of the specific application of the sensor constructed by the polystyrene matrix composite with 3D graphitic hollow shells. What fields and scenarios can it be used in?

Author Response

Please see the attached response file

Reviewer 2 Report

Comments and Suggestions for Authors

This manuscript presents interesting results, which are useful for constructing sensitive and accurate piezoresistive sensors. There are a few questions to be addressed:

a)      The dependence of the electrical conductivity as a function of filler content is depicted. Which value of the filler content is best for optimum functionality of the piezoeresistive sensors and why? Compare with findings in Materials Chemistry and Physics  232, 319-324 (2019), whereas, the carbon content should be just above the percolation threshold (~1%) so as the polymer/nano-carbon composite exhibit best electro-switching performance upon compression and include comments in your text.

b)      Table 2: You must keep only significant digits, dictated by the error values. For example, 0.223±0.034, should be weitten by keeping (experimentally/statistically) significant digitsQ 0.22±0.03 .

Author Response

Please see the attached response file

Round 2

Reviewer 1 Report

Comments and Suggestions for Authors

In the revised manuscript, the authors have addressed my concerns with the original one. It could be accepted after minor editing of the English language and  text.

Comments on the Quality of English Language

The English language needs to be edited slightly